# Chefs' Random Tables: Non-Trigonometric Random Features

**Valerii Likhosherstov***
University of Cambridge
vl304@cam.ac.uk

**Krzysztof Choromanski***
Google Research & Columbia University
kchoro@google.com

**Avinava Dubey***
Google Research

**Frederick Liu***
Google Research

**Tamas Sarlos**
Google Research

**Adrian Weller**
University of Cambridge &
The Alan Turing Institute

## Abstract

We introduce *chefs' random tables* (CRTs), a new class of non-trigonometric random features (RFs) to approximate Gaussian and softmax-kernels. CRTs are an alternative to standard random kitchen sink (RKS) methods, which inherently rely on the trigonometric maps [46]. We present variants of CRTs where RFs are positive, a key requirement for applications in recent low-rank Transformers [15]. Further variance reduction is possible by leveraging statistics which are simple to compute. One instantiation of CRTs, the *optimal positive random features* (OPRFs), is to our knowledge the first RF method for unbiased softmax-kernel estimation with positive and bounded RFs, resulting in exponentially small tails and much lower variance than its counterparts. As we show, orthogonal random features applied in OPRFs provide additional variance reduction for any dimensionality $d$ (not only asymptotically for sufficiently large $d$, as for RKS). We test CRTs on many tasks ranging from non-parametric classification to training Transformers for text, speech and image data, obtaining new state-of-the-art results for low-rank text Transformers, while providing linear space and time complexity of the attention.

## 1 Introduction & related work

The idea that nonlinear mappings of the random-weight linear combinations of data features can be used to linearize various nonlinear similarity functions transformed kernel methods. This led to the development of *Random Kitchen Sinks* (RKSs) techniques; and the new field of scalable kernel algorithms, introduced in the paper trilogy [44, 45, 46], was born. RKSs were subsequently used in many applications, ranging from kernel and function-to-function regression [1, 33, 41], SVM algorithms [50] to operator-valued and semigroup kernels [36, 62], neural networks [25, 61, 11, 27] and even differentially-private ML algorithms [9], as well as (very recently) nonparametric adaptive control [4]. Random features (RFs) are a subject of much theoretical analysis [34, 63, 51, 48].

To approximate shift invariant (e.g. Gaussian, Cauchy or Laplace) and softmax kernels, RKSs rely on the trigonometric nonlinear mappings provided directly by Bochner's Theorem [36]. Trigonometric RFs provide strong concentration results (e.g. uniform convergence, see Claim 1 in [45]), but suffer from a weakness that was noted recently – they are not guaranteed to be positive. This makes them unsuitable for approximating softmax-attention in scalable Transformers relying on implicit attention via random features [15]. As noted in [15], trigonometric features lead to unstable training, as they yield poor approximations of the partition functions applied to renormalize attention and involving

---

* Equal contribution

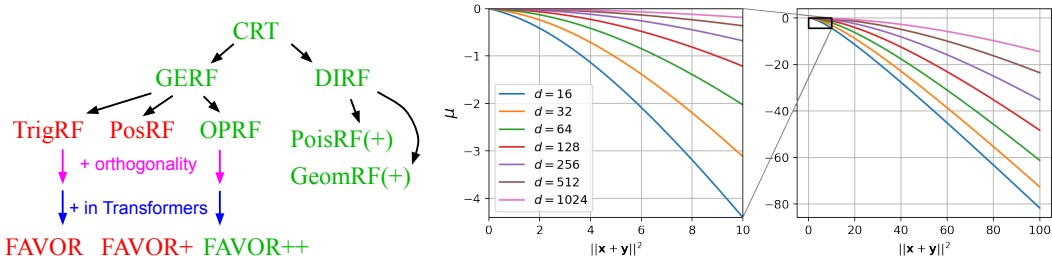

Figure 1: **(left)** A map of RF methods for the Gaussian kernel approximation. Existing RFs (Section 2.2), RFs proposed in this paper. **(right)** The utility function $\mu$ (defined as the logarithm of the ratio of the variance of OPRF and PosRF mechanisms for the Gaussian and softmax kernel estimation) as a function of squared length of the sum of kernels' inputs $\|\mathbf{x} + \mathbf{y}\|^2$ (smaller values imply larger gains coming from OPRF). Different curves correspond to different dimensionalities. Based on the plots, OPRFs have $> e^{60}$ times smaller variance when $d = 64, \|\mathbf{x} + \mathbf{y}\|^2 = 100$ (configuration taken from the standard Transformer application).

several small softmax kernel values. To address this, [15] proposed a new method for unbiased softmax kernel estimation with positive RFs, the so-called *FAVOR+* mechanism (**F**ast **A**ttention **V**ia **O**rthogonal **R**andom Positive Features), as opposed to FAVOR using trigonometric RFs (as in [13]). Positive random features guarantee that the denominator in self-attention is a sum of positive numbers, hence it cannot be negative or too small.

Unfortunately FAVOR+ features are not bounded, and worse, the moment generating function of the corresponding softmax kernel estimator is not finite. Consequently, no concentration results beyond those involving second moment methods (variance bounds) have been provided for FAVOR+. Despite active research on constructing new RFs for implicit attention in Transformers [12, 17], the following questions of great practical importance remained open:

*Does there exist an unbiased estimator of the softmax/Gaussian kernel relying on positive and simultaneously bounded random features? Can it be efficiently constructed?*

We answer both questions affirmatively in this paper, introducing a new mechanism called *optimal positive random features* (OPRFs). We propose other RF methods that, as OPRFs, do not apply trigonometric functions and provide positivity. We call this new set of RF mechanisms *chefs' random tables* (CRTs, see Figure 1-left). The new OPRF-based method for fast self-attention approximation, applying in addition block-orthogonal random projections, is referred to as *FAVOR++*.

We compute the variance of OPRF-based estimators (see Theorem 3.1 & 3.2) and show that they can provide $e^{60}$x variance reduction for Gaussian/softmax kernel estimation (see Figure 1). We give the first exponentially small upper bounds for tails of the Gaussian/softmax kernel estimators relying on positive RFs, leveraging boundedness of OPRFs (see Theorem 4.2). Consequently, using OPRFs we give the first uniform convergence results for softmax attention approximation with positive RFs (Theorem 4.3). We show that orthogonal random projections combined with OPRFs (leading to FAVOR++) provably reduce the variance of OPRFs for any dimensionality $d$ (see Theorem 4.1) as opposed to only asymptotically for $d$ large enough which is the case for RKSs. Finally, we provide extensive empirical evaluation in Section 5, for Transformers (text, image and speech domains), establishing new state-of-the-art results for low-rank attention Transformers for text.

## 2 Prerequisites

### 2.1 The definition of random features

Let $\mathbf{x}, \mathbf{y} \in \mathbb{R}^d$ be real vectors and $K(\mathbf{x}, \mathbf{y}) = \exp(-\frac{1}{2}\|\mathbf{x} - \mathbf{y}\|^2)$ be a Gaussian kernel where $\|\cdot\|$ denotes the $L_2$-norm. By *random features (RFs) for the Gaussian kernel* we denote two functions $f^{(1)}(\omega, \mathbf{x}), f^{(2)}(\omega, \mathbf{y}) : \mathbb{R}^d \times \mathbb{R}^d \to \mathbb{C}$ where $\omega$ is a random vector from some distribution $p(\omega)$ on $\mathbb{R}^d$. Functions $f^{(\cdot)}(\omega, \mathbf{x})$ satisfy the following:

$$K(\mathbf{x}, \mathbf{y}) = \mathbb{E}_{p(\omega)}\text{Re}\left(f^{(1)}(\omega, \mathbf{x})f^{(2)}(\omega, \mathbf{y})\right) \tag{1}$$

for all $\mathbf{x}, \mathbf{y} \in \mathbb{R}^d$ where $\text{Re}(\cdot)$ denote the real part of a complex number ($\text{Im}(\cdot)$ for the imaginary part). The decomposition (1) can be used for an unbiased approximation of the linear operator

$\mathcal{K} = (K(\mathbf{x}^{(i)}, \mathbf{y}^{(j)}))_{i,j=1}^{L,L} \in \mathbb{R}^{L \times L}$ where $\mathbf{x}^{(i)}, \mathbf{y}^{(j)} \in \mathbb{R}^d$. Such linear operators emerge in various applications, e.g. kernel SVM [45], kernel regression [38, 59] or Transformers [15] (see Section 2.3).

For any $\mathbf{c} \in \mathbb{R}^L$, evaluating $\mathcal{K}\mathbf{c}$ naively would result in $O(dL^2)$ time complexity which is prohibitively expensive for large $L$. Instead, we can use the Monte Carlo approximation: draw i.i.d. samples $\omega_1, \ldots, \omega_M \sim p(\omega)$, where $M \ll L$, and compute for $1 \leq i \leq L$:

$$(\mathcal{K}\mathbf{c})_i = \sum_{j=1}^{L} K(\mathbf{x}^{(i)}, \mathbf{y}^{(j)})\mathbf{c}_j \approx \sum_{j=1}^{L} \left( \frac{1}{M} \sum_{m=1}^{M} \mathrm{Re} \left( f^{(1)}(\omega_m, \mathbf{x}^{(i)}) f^{(2)}(\omega_m, \mathbf{y}^{(j)}) \right) \right) \mathbf{c}_j$$

$$= \frac{1}{M} \mathrm{Re} \left( \sum_{m=1}^{M} f^{(1)}(\omega_m, \mathbf{x}^{(i)}) \sum_{j=1}^{L} f^{(2)}(\omega_m, \mathbf{y}^{(j)})\mathbf{c}_j \right). \tag{2}$$

Therefore, $\mathcal{K}\mathbf{c}$ can be approximated by first precomputing $\{\sum_{j=1}^{L} f^{(2)}(\omega_m, \mathbf{y}^{(j)})\mathbf{c}_j\}_{m=1}^{M}$ and then evaluating (2) in $O(dML)$ total time. Precision of this approximation can be theoretically bounded [45, 15]. In this manuscript, we will use the variance $\mathrm{Var}_{p(\omega)} \mathrm{Re} \left( f^{(1)}(\omega, \mathbf{x}) f^{(2)}(\omega, \mathbf{y}) \right)$ of (1) to judge the precision of the Monte Carlo approximation. Since samples $\omega_1, \ldots, \omega_M$ are i.i.d., the number $M$ of RFs controls the tradeoff between the total variance $\mathrm{Var}_{\omega_1,\ldots,\omega_M}(\ldots) = \frac{1}{M} \mathrm{Var}_{p(\omega)}(\ldots)$ (inversely proportional to $M$) and number of computations (directly proportional to $M$).

The softmax kernel is defined as: $K_{\mathrm{sfm}}(\mathbf{x}, \mathbf{y}) = \exp(\mathbf{x}^T \mathbf{y})$, and can be easily derived from the Gaussian kernel $K$ as follows: $K_{\mathrm{sfm}}(\mathbf{x}, \mathbf{y}) = \exp(\|\mathbf{x}\|^2/2) K(\mathbf{x}, \mathbf{y}) \exp(\|\mathbf{y}\|^2/2)$. Thus, in particular, any RF mechanism for the Gaussian kernel immediately transfers to the corresponding one for the softmax kernel and vice versa. Thus from now on, unless explicitly stated otherwise, the estimators we consider are approximating the Gaussian kernel.

## 2.2 Existing trigonometric and positive random feature methods

Here we summarize existing RFs for Gaussian kernel estimation. *Trigonometric RFs (TrigRFs)*, the core of RKSs [45] and FAVOR [13], are defined as follows: $f_{\mathrm{trig}}^{(1)}(\omega, \mathbf{x}) = \exp(\mathrm{i}\omega^\top \mathbf{x})$, $f_{\mathrm{trig}}^{(2)}(\omega, \mathbf{y}) = \exp(-\mathrm{i}\omega^\top \mathbf{y})$, $p_{\mathrm{trig}}(\omega) \sim \mathcal{N}(\mathbf{0}_d, \mathbf{I}_d)$ where i denotes an imaginary unit (as opposed to the index notation $i$), $\mathbf{0}_d \in \mathbb{R}^d$ is a vector of zeros and $\mathbf{I}_d \in \mathbb{R}^{d \times d}$ is an identity matrix. The variance of these RFs has the following form [15]: $\mathrm{Var}_{p_{\mathrm{trig}}(\omega)} \mathrm{Re} \left( f_{\mathrm{trig}}^{(1)}(\omega, \mathbf{x}) f_{\mathrm{trig}}^{(2)}(\omega, \mathbf{y}) \right) = \frac{1}{2} \left( 1 - K(\mathbf{x}, \mathbf{y})^2 \right)^2$.

*Positive RFs (PosRFs)* [15], the key ingredient of the FAVOR+ mechanism, are defined as follows: $f_{\mathrm{pos}}^{(1)}(\omega, \mathbf{x}) = f_{\mathrm{pos}}^{(2)}(\omega, \mathbf{x}) = \exp(\omega^\top \mathbf{x} - \|\mathbf{x}\|^2)$, $p_{\mathrm{pos}}(\omega) \sim \mathcal{N}(\mathbf{0}_d, \mathbf{I}_d)$. Their name is due to the fact that $f_{\mathrm{pos}}^{(1)}(\omega, \mathbf{x}), f_{\mathrm{pos}}^{(2)}(\omega, \mathbf{y})$ are always positive real numbers. PosRF variance has the form [15]: $\mathrm{Var}_{p_{\mathrm{pos}}(\omega)} \left( f_{\mathrm{pos}}^{(1)}(\omega, \mathbf{x}) f_{\mathrm{pos}}^{(2)}(\omega, \mathbf{y}) \right) = \exp(4\mathbf{x}^\top \mathbf{y}) - K(\mathbf{x}, \mathbf{y})^2$.

## 2.3 Random features for scalable Transformers

One recent application of RFs is in the area of scalable Transformers for processing long sequences [15]. Let $L$ be the length of the sequence. Interactions between elements in Transformers are implemented via the *self-attention mechanism*. Given three matrices $\mathbf{Q}, \mathbf{K}, \mathbf{V} \in \mathbb{R}^{L \times d}$, the self-attention mechanism returns the following result:

$$\mathbf{Y} = \mathrm{softmax}(d^{-1/2} \mathbf{Q}\mathbf{K}^\top)\mathbf{V} = \mathrm{diag}(\mathcal{K}_{\mathrm{sfm}}\mathbf{1}_L)^{-1} \mathcal{K}_{\mathrm{sfm}}\mathbf{V}, \quad \mathcal{K}_{\mathrm{sfm}} = (K_{\mathrm{sfm}}(\mathbf{x}_i, \mathbf{y}_j))_{i,j=1}^{L,L} \tag{3}$$

where $\mathbf{1}_L \in \mathbb{R}^L$ is a vector of all ones, $\mathbf{x}^{(i)} = d^{-1/4} \mathbf{Q}_{i,:}$ ($i$'th row of $\mathbf{Q}$) and $\mathbf{y}^{(j)} = d^{-1/4} \mathbf{K}_{j,:}$, $1 \leq i, j \leq L$. We deduce that computing (3) reduces to applying the linear operator $\mathcal{K}_{\mathrm{sfm}}$ to $d + 1$ vectors: $\mathbf{1}_L, \mathbf{V}_{:,1}, \ldots, \mathbf{V}_{:,d}$. Hence, when $L$ is large, RF approximation similar to (2) but for the $K_{\mathrm{sfm}}(\cdot, \cdot)$ kernel can reduce the computational complexity from $O(dL^2)$ to $O(dML)$.

Importantly, when the approximation (2) with the replacement $\mathcal{K} \to \mathcal{K}_{\mathrm{sfm}}$ can take negative and/or near-zero values, training is unstable since this approximation emerges in the denominator (inversed) term $\mathrm{diag}(\mathcal{K}_{\mathrm{sfm}}\mathbf{1}_L)$ in (3). One way to address this is to restrict $f^{(1)}(\omega, \mathbf{x})$ and $f^{(2)}(\omega, \mathbf{y})$ to always map into strictly positive numbers $\mathbb{R}^+$. This is where PosRFs introduced in 2.2 are particularly relevant.

# 3 Chefs' Random Tables

We are ready to present our mechanism of chefs' random tables. All proofs are in the Appendix.

## 3.1 Generalized exponential RFs (GERFs) & optimal positive RFs (OPRFs)

Our first goal will be to generalize both trigonometric and positive RFs. Then we will focus on one special case of this generalization, that will directly lead to the FAVOR++ mechanism.

We will be looking for RFs of the following generalized exponential form for $p_{\mathrm{GE}}(\omega) \sim \mathcal{N}(\mathbf{0}_d, \mathbf{I}_d)$:

$$
\begin{aligned}
f_{\mathrm{GE}}^{(1)}(\omega, \mathbf{x}) &= D \exp(A\|\omega\|^2 + B\omega^\top \mathbf{x} + C\|\mathbf{x}\|^2), \\
f_{\mathrm{GE}}^{(2)}(\omega, \mathbf{y}) &= D \exp(A\|\omega\|^2 + sB\omega^\top \mathbf{y} + C\|\mathbf{y}\|^2),
\end{aligned}
\tag{4}
$$

where $A, B, C, D \in \mathbb{C}$ and $s \in \{-1, +1\}$. It can be seen that $A = 0$, $B = \mathrm{i}$, $C = 0$, $D = 1$, $s = -1$ corresponds to trigonometric RFs and $A = 0$, $B = 1$, $C = -1$, $D = 1$, $s = 1$ corresponds to positive RFs. The next theorem describes the conditions under which $f_{\mathrm{GE}}^{(\cdot)}$ can be used to approximate the Gaussian kernel.

**Theorem 3.1.** $p_{\mathrm{GE}}(\omega)$ and $f_{\mathrm{GE}}^{(\cdot)}$, defined in (4), satisfy (1) if

$$
\mathrm{Re}\,(1 - 4A) > 0, \quad B = \sqrt{s(1 - 4A)}, \quad C = -(s+1)/2, \quad D = (\sqrt[4]{1 - 4A})^d,
\tag{5}
$$

where $\sqrt{\cdot}$ and $\sqrt[n]{\cdot}$ denotes a principal root if the argument is complex.

Hence, $A$ and $s$ can be treated as free parameters and $B, C, D$ as dependent ones. The variance of these RFs can be expressed through $A$ and $s$ as follows:

**Theorem 3.2.** Let $\mathrm{Re}\,(1 - 8A) > 0$. The variance of (2) using $p_{\mathrm{GE}}(\omega)$, $f_{\mathrm{GE}}^{(\cdot)}$ is given as

$$
\begin{aligned}
\mathrm{Var}_{p_{\mathrm{GE}}(\omega)} \mathrm{Re}\left(f_{\mathrm{GE}}^{(1)}(\omega, \mathbf{x}) f_{\mathrm{GE}}^{(2)}(\omega, \mathbf{y})\right) &= \frac{1}{2} \exp\left(-(s+1)\left(\|\mathbf{x}\|^2 + \|\mathbf{y}\|^2\right)\right) \\
&\times \left(\mathrm{Re}\left(\alpha_1 \exp\left(\alpha_2 \|\mathbf{x} + s\mathbf{y}\|^2\right)\right) + \alpha_3 \exp\left(\alpha_4 \|\mathbf{x} + s\mathbf{y}\|^2\right)\right) - K(\mathbf{x}, \mathbf{y})^2.
\end{aligned}
\tag{6}
$$

where $\alpha_1 = \left(\sqrt{1 + \frac{16A^2}{1-8A}}\right)^d$, $\alpha_2 = \left(s + \frac{s}{1-8A}\right)$, $\alpha_3 = \left(1 + \frac{16|A|^2}{1-8\mathrm{Re}(A)}\right)^{d/2}$, $\alpha_4 = \left(\frac{s}{2} + \frac{s+2|1-4A|}{2(1-8\mathrm{Re}(A))}\right)$.

While it is unclear how to find a global minimum of the objective (6) with respect to $A \in \mathbb{C}$, $\mathrm{Re}\,(1 - 8A) > 0$ and $s \in \{-1, +1\}$, we observe that it's possible to find an optimum when we restrict $A$ to be a real number and fix $s = +1$.

**Theorem 3.3** (Minimum variance). *When $s = +1$, $A$ is restricted to be a real number and $\|\mathbf{x}+\mathbf{y}\|^2 > 0$, the variance (6) is minimized when $A = (1 - 1/\rho^*)/8$ where $0 < \rho^* < 1$,*

$$
\rho^* = \left(\sqrt{(2\|\mathbf{x} + \mathbf{y}\|^2 + d)^2 + 8d\|\mathbf{x} + \mathbf{y}\|^2} - 2\|\mathbf{x} + \mathbf{y}\|^2 - d\right) / \left(4\|\mathbf{x} + \mathbf{y}\|^2\right).
\tag{7}
$$

**Note:** One can show that $A < 0$ for $\mathbf{x} \neq -\mathbf{y}$ thus the corresponding estimator is bounded since the term $A\|\omega\|^2$ prevails over linear terms $B\omega^\top \mathbf{x}$ and $sB\omega^\top \mathbf{y}$ in (4). When $\|\mathbf{x} + \mathbf{y}\| \to 0$ then $\rho^* \to 1$ and thus $A \to 0$. Therefore for $\mathbf{x} = -\mathbf{y}$ the mechanism reduces to PosRF described in Sec. 2.2 as expected, since for $\mathbf{x} = -\mathbf{y}$ PosRFs provide perfect estimation (variance equal to zero). Larger values of $\|\mathbf{x} + \mathbf{y}\|$ lead to larger gains coming from the new mechanism.

From (5) it can be inferred that $B, C, D$ are real when $A$ is real and $s = +1$. Hence, $f^{(1)}(\omega, \mathbf{x})$, $f^{(2)}(\omega, \mathbf{y})$ are positive real numbers in this case. Furthermore, $s = +1$, $A = 0$ corresponds to positive RFs. Therefore, we refer to RFs with $A$ defined according to (7) as *optimal positive RFs* (OPRFs). Figure 1-right illustrates the analytical variance reduction achieved via OPRFs.

In practice, we are given sets $\{\mathbf{x}^{(i)}\}$, $\{\mathbf{y}^{(j)}\}$ instead of a single pair $\mathbf{x}, \mathbf{y}$. For this reason, in (6,7), we can use the averages of $\|\mathbf{x}^{(i)}\|^2$, $\|\mathbf{y}^{(j)}\|^2$, $\|\mathbf{x}^{(i)} + s\mathbf{y}^{(j)}\|^2$ instead of $\|\mathbf{x}\|^2$, $\|\mathbf{y}\|^2$, $\|\mathbf{x} + s\mathbf{y}\|^2$. This heuristic is based on the assumption that all $\{\mathbf{x}^{(i)}\}$ and $\{\mathbf{y}^{(j)}\}$ are homogeneous and $\|\mathbf{x}^{(i)}\|^2$,

$\|\mathbf{y}^{(j)}\|^2$, $\|\mathbf{x}^{(i)} + s\mathbf{y}^{(j)}\|^2$ are tightly concentrated around their mean. Computing averages of $\|\mathbf{x}^{(i)}\|^2$, $\|\mathbf{y}^{(j)}\|^2$ takes $O(Ld)$ time. Using the formula below, the average of $\|\mathbf{x}^{(i)} + s\mathbf{y}^{(j)}\|^2$ can be computed with the same complexity:

$$\frac{1}{L^2} \sum_{i=1}^{L} \sum_{j=1}^{L} \|\mathbf{x}^{(i)} + s\mathbf{y}^{(j)}\|^2 = \frac{1}{L} \sum_{i=1}^{L} \|\mathbf{x}^{(i)}\|^2 + \frac{2s}{L^2} \left( \sum_{i=1}^{L} \mathbf{x}^{(i)} \right)^{\top} \left( \sum_{i=1}^{L} \mathbf{y}^{(i)} \right) + \frac{1}{L} \sum_{i=1}^{L} \|\mathbf{y}^{(i)}\|^2. \quad (8)$$

The closed-form solution for real $A$ and $s = +1$ allows $O(1)$-time optimization of (6) after precomputing these statistics. In the general case we can rely on numerical optimization of (6) with respect to $A \in \mathbb{C}$ and $s \in \{-1, +1\}$. Using precomputed statistics, each evaluation of (6) takes $O(1)$ time. As long as the total number of these evaluations is $O(LM(d + n))$, where $n$ is the number of $\mathcal{K}$ or $\mathcal{K}_{\mathrm{sfm}}$ evaluations ($n = d + 1$ in Section 2.3), it does not affect the total complexity.

The next class of mechanisms, if implemented straightforwardly, does not give positive-valued RFs but, as we explain in Section 3.2.3, can be easily transformed to variants providing positivity.

## 3.2 Discretely-induced random features (DIRFs)

Take a discrete probabilistic distribution $p(\omega)$ where $\omega_1, \ldots, \omega_d$ are i.i.d. with $\mathbb{P}(\omega_l = k) = p_k$, $\sum_{k=0}^{\infty} p_k = 1$ and $p_k > 0$ for $k \in \{0\} \cup \mathbb{N}$. Note that, by Taylor series expansion of $\exp(\cdot)$,

$$K(\mathbf{x}, \mathbf{y}) \exp(\frac{\|\mathbf{x}\|^2}{2}) \exp(\frac{\|\mathbf{y}\|^2}{2}) = \exp(\mathbf{x}^{\top} \mathbf{y}) = \prod_{l=1}^{d} \sum_{k=0}^{\infty} p_k \frac{\mathbf{x}_l^k \mathbf{y}_l^k}{p_k k!} = \mathbb{E} \left[ \prod_{l=1}^{d} X_l \prod_{l=1}^{d} Y_l \right], \quad (9)$$

where $X_l = \mathbf{x}_l^{\omega_l} (\omega_l!)^{-\frac{1}{2}} p_{\omega_l}^{-\frac{1}{2}}$, $Y_l = \mathbf{y}_l^{\omega_l} (\omega_l!)^{-\frac{1}{2}} p_{\omega_l}^{-\frac{1}{2}}$. Thus we can define *discretely-induced random features* providing Gaussian kernel estimation as follows:

$$f_{\mathrm{DI}}^{(1)}(\omega, \mathbf{x}) = f_{\mathrm{DI}}^{(2)}(\omega, \mathbf{x}) = f_{\mathrm{DI}}(\omega, \mathbf{x}) = \exp(-\frac{\|\mathbf{x}\|^2}{2}) \prod_{l=1}^{d} x_i^{\omega_l} (\omega_l!)^{-\frac{1}{2}} p_{\omega_l}^{-\frac{1}{2}}. \quad (10)$$

Different instantiations of the above mechanism are given by different probabilistic distributions $\{p_k\}$. We will consider two prominent special cases: (a) Poisson, and (b) geometric distributions.

### 3.2.1 Poisson random features (PoisRFs)

If $\{p_k\}$ is a Poisson distribution, i.e. $p_k = e^{-\lambda} \lambda^k / k!$, $k \in \{0\} \cup \mathbb{N}$, then the corresponding RFs are defined as: $f_{\mathrm{pois}}^{(1)}(\omega, \mathbf{x}) = f_{\mathrm{pois}}^{(2)}(\omega, \mathbf{x}) = f_{\mathrm{pois}}(\omega, \mathbf{x}) = e^{\lambda d/2 - \|\mathbf{x}\|^2/2} \prod_{l=1}^{d} \mathbf{x}_l^{\omega_l} \lambda^{-\omega_l/2}$.

**Theorem 3.4.** *Variance of (2) with $p_{\mathrm{pois}}, f_{\mathrm{pois}}$ is given by*

$$\mathrm{Var}_{p_{\mathrm{pois}}(\omega)} (f_{\mathrm{pois}}(\omega, \mathbf{x}) f_{\mathrm{pois}}(\omega, \mathbf{y})) = \exp\left( \lambda d + \lambda^{-1} \sum_{l=1}^{d} \mathbf{x}_l^2 \mathbf{y}_l^2 - \|\mathbf{x}\|^2 - \|\mathbf{y}\|^2 \right) - K(\mathbf{x}, \mathbf{y})^2. \quad (11)$$

The $\exp$ argument in (11) is convex as a function of $\lambda > 0$. By setting its derivative to zero, we find that $\lambda^* = d^{-1/2} (\sum_{l=1}^{d} \mathbf{x}_l^2 \mathbf{y}_l^2)^{1/2}$ gives the minimum of (11).

When, instead of a single pair $\mathbf{x}, \mathbf{y}$, sets $\{\mathbf{x}^{(i)}\}$, $\{\mathbf{y}^{(j)}\}$ are provided, we can use the same homogeneity assumption as in Section 3.1 and substitute the average of $\sum_{l=1}^{d} (\mathbf{x}_l^{(i)})^2 (\mathbf{y}_l^{(j)})^2$ over $1 \le i, j \le L$ instead of $\sum_{l=1}^{d} \mathbf{x}_l^2 \mathbf{y}_l^2$. This average can be computed efficiently in $O(Ld)$ time as follows:

$$L^{-2} \sum_{i=1}^{L} \sum_{j=1}^{L} \sum_{l=1}^{d} (\mathbf{x}_l^{(i)})^2 (\mathbf{y}_l^{(j)})^2 = L^{-2} \sum_{l=1}^{d} \left( \sum_{i=1}^{L} (\mathbf{x}_l^{(i)})^2 \right) \left( \sum_{i=1}^{L} (\mathbf{y}_l^{(i)})^2 \right). \quad (12)$$

After computing this statistic, we can calculate $\lambda^*$ in $O(1)$ time using the analytic formula.

### 3.2.2 Geometric random features (GeomRFs)

If $\{p_k\}$ is a geometric distribution, i.e. $p_k = p(1-p)^k$, $k \in \{0\} \cup \mathbb{N}$, for a parameter $0 < p < 1$, then the corresponding RFs are defined as: $f_{\text{geom}}^{(1)}(\omega, \mathbf{x}) = f_{\text{geom}}^{(2)}(\omega, \mathbf{x}) = f_{\text{geom}}(\omega, \mathbf{x}) = p^{-d/2}e^{-\|\mathbf{x}\|^2/2}\prod_{l=1}^{d}\mathbf{x}_l^{\omega_l}(1-p)^{-\omega_l/2}(\omega_l!)^{-1/2}$.

**Theorem 3.5.** *The variance of (2) with $p_{\text{geom}}, f_{\text{geom}}$ is given as*

$$\text{Var}_{p_{\text{geom}}(\omega)}(f_{\text{geom}}(\omega,\mathbf{x})f_{\text{geom}}(\omega,\mathbf{y})) = p^{-d}e^{-\|\mathbf{x}\|^2-\|\mathbf{y}\|^2}\prod_{l=1}^{d}I_0(2(1-p)^{-\frac{1}{2}}|\mathbf{x}_l\mathbf{y}_l|) - K(\mathbf{x},\mathbf{y})^2 \quad (13)$$

*where $I_0(\cdot)$ is the modified Bessel function of the first kind of order $0$.*

Again as for the previously described mechanisms, when sets $\{\mathbf{x}^{(i)}\}$, $\{\mathbf{y}^{(j)}\}$ are given, we can use averages of $|\mathbf{x}_l^{(i)}\mathbf{y}_l^{(j)}|$, $1 \le l \le d$, instead of $|\mathbf{x}_l\mathbf{y}_l|$ in (13) assuming homogeneity of $\mathbf{x}^{(i)}$'s and $\mathbf{y}^{(j)}$'s. Each out of $d$ averages can be computed in $O(L)$ time as follows:

$$L^{-2}\sum_{i=1}^{L}\sum_{j=1}^{L}|\mathbf{x}_l^{(i)}\mathbf{y}_l^{(j)}| = L^{-2}\left(\sum_{i=1}^{L}|\mathbf{x}_l^{(i)}|\right)\left(\sum_{i=1}^{L}|\mathbf{y}_l^{(i)}|\right). \quad (14)$$

After precomputation of these statistics, evaluation of (11) takes $O(d)$ time. A numerical optimization can be used to minimize (11) with respect to $p$. As long as the number of variance evaluations is $O(LM(1 + n/d))$, the total complexity estimate is not affected.

### 3.2.3 Making discretely-induced RFs positive

As can be inferred from Eq. 10, DIRFs are positive when all elements of $\mathbf{x}$ and $\mathbf{y}$ are positive. If this is not the case, and positive-valued RFs are needed, e.g. in applications involving scalable Transformers, one way to make them positive is to take some vector $\mathbf{c} \in \mathbb{R}^d$ such that $\mathbf{c}_l < \mathbf{x}_l, \mathbf{y}_l$. An example of such a vector is given by $c_l = \min_i \min(\mathbf{x}_l^{(i)}, \mathbf{y}_l^{(i)}) - \epsilon$ where $\epsilon > 0$ is a small constant. Next, define $\widehat{\mathbf{x}}^{(i)} = \mathbf{x}^{(i)} - \mathbf{c}$, $\widehat{\mathbf{y}}^{(j)} = \mathbf{y}^{(j)} - \mathbf{c}$. Then, clearly, $\widehat{\mathbf{x}}^{(i)} - \widehat{\mathbf{y}}^{(j)} = \mathbf{x}^{(i)} - \mathbf{y}^{(j)}$, $K(\widehat{\mathbf{x}}^{(i)}, \widehat{\mathbf{y}}^{(j)}) = K(\mathbf{x}^{(i)}, \mathbf{y}^{(j)})$ and RFs can be used on $\widehat{\mathbf{x}}^{(i)}, \widehat{\mathbf{y}}^{(j)}$ which have positive entries. We refer to these variants of PoisRFs and GeomRFs as PoisRF+ and GeomRF+ respectively.

## 4 Additional theoretical results & FAVOR++

Interestingly, as in the case of the PosRF mechanism from [15], OPRFs also benefit from applying block-orthogonal ensembles of projections $\omega$ (see Appendix 9.1 and [15] for the exact definition). We show below that orthogonal RFs reduce the variance of GERFs with $A \in \mathbb{R}$, $s = 1$ (which is the case for OPRF) for any $d > 0$:

**Theorem 4.1** (Orthogonal OPRFs). *If $\text{Var}(\widehat{K}_M^{\text{ort}}(\mathbf{x}, \mathbf{y}))$ denotes the variance of the orthogonal GERF estimator $\widehat{K}_M^{\text{ort}}(\mathbf{x}, \mathbf{y})$ of the Gaussian kernel with $A \in \mathbb{R}, s = 1$ in (4) at $\mathbf{x}, \mathbf{y} \in \mathbb{R}^d$ using $M$ RFs and $\text{Var}(\widehat{K}_M^{\text{iid}}(\mathbf{x}, \mathbf{y}))$ is the analogous expression with i.i.d. samples, then for some $\mathcal{C}(\|\mathbf{x} + \mathbf{y}\|) \ge 0$:*

$$\text{Var}(\widehat{K}_M^{\text{ort}}(\mathbf{x}, \mathbf{y})) \le \text{Var}(\widehat{K}_M^{\text{iid}}(\mathbf{x}, \mathbf{y})) - (1 - \frac{1}{M})\frac{2}{d+2}\mathcal{C}(\|\mathbf{x} + \mathbf{y}\|). \quad (15)$$

**Note:** The analogous inequality can be obtained for TrigRFs only in the asymptotic sense (for $d$ large enough, see Theorem 3.8 in [14]). One of the key properties used in the proof of Theorem 4.1 is positivity of RFs. We conclude that positive-valued RFs are particularly well suited for the quasi Monte-Carlo methods based on the orthogonal ensembles. Analogously to FAVOR+ [15], we refer to the self-attention approximation mechanism based on orthogonal OPRFs as *FAVOR++*.

We now provide strong concentration results for the OPRF-based estimators critically relying on the boundedness of OPRFs. Boundedness of OPRFs is due to $A = (1 - 1/\rho^*)/8 < 0$ even when $\|\mathbf{x} + \mathbf{y}\|^2$ is substituted by average (8) in (7). To the best of our knowledge, these are the first such results for positive-valued RFs. Denote by $\mathcal{L}$ the Legendre Transform of the random variable $Z = f_{\text{GE}}^{(1)}(\omega, \mathbf{x})f_{\text{GE}}^{(2)}(\omega, \mathbf{y})$ for $f_{\text{GE}}^{(\cdot)}$ as in (4) with $A, B, C, D \in \mathbb{R}$ defining GERFs.

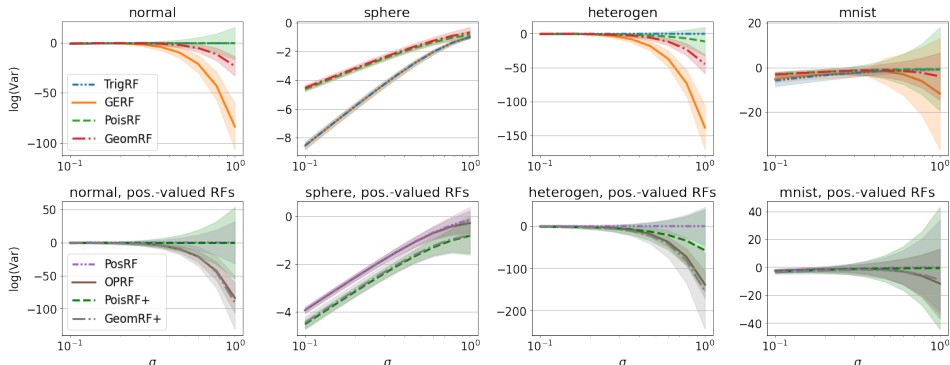

Figure 2: Log-variance of different RF mechanisms, mean and standard deviation. For each sampling method, we plot the results for non-positive and positive RFs on separate plots for $0.1 \leq \sigma \leq 1$.

**Theorem 4.2.** *Suppose $A < 0$. The following is true for any $\epsilon > 0$:* $\mathbb{P}[|\widehat{K}_M^{\mathrm{iid}}(\mathbf{x}, \mathbf{y}) - K(\mathbf{x}, \mathbf{y})| \geq \epsilon] \leq 2\exp(-\frac{M\epsilon^2}{2}\exp(\frac{\|\mathbf{x}\|^2 + \|\mathbf{y}\|^2}{2A})(1 - 4A)^{-\frac{d}{2}})$. *Furthermore, for the orthogonal variant we have:* $\mathbb{P}[\widehat{K}_M^{\mathrm{ort}}(\mathbf{x}, \mathbf{y}) - K(\mathbf{x}, \mathbf{y}) \geq \epsilon] \leq \exp(-M\mathcal{L}(K(\mathbf{x}, \mathbf{y}) + \epsilon))$ *and* $\mathcal{L}(K(\mathbf{x}, \mathbf{y}) + \epsilon)) > 0$.

Finally, below we provide the first result regarding uniform convergence for attention approximation in the efficient low-rank Transformers (Section 2.3).

**Theorem 4.3** (Uniform convergence for attention approximation). *Assume that rows of $\mathbf{Q}$ and $\mathbf{K}$ from (3) come from the $L_2$-ball of radius $R > 0$. Denote by $\widehat{\mathcal{K}}_{\mathrm{sfm}}$ the approximation of $\mathcal{K}_{\mathrm{sfm}}$ from (3) via the GERF-mechanism with $A < 0$ using $M$ independent random projections. Then $\|\mathcal{K}_{\mathrm{sfm}} - \widehat{\mathcal{K}}_{\mathrm{sfm}}\|_\infty \leq \epsilon$ with any constant probability when $M = \Omega((1 - 4A)^{\frac{d}{2}}\Gamma\frac{d}{\epsilon^2}\log(\frac{\gamma\rho}{\epsilon}))$, where: $\Gamma = \exp(-\frac{3R^2}{\sqrt{d}A})$, $\rho = \sqrt{2}Rd^{-\frac{1}{4}}$, $\gamma = (1 - 4A)^{\frac{d}{2}}\sqrt{4\Gamma(\frac{R^2}{\sqrt{d}} + d^2)}$ (for $A$ as in the OPRFs definition).*

## 5 Experiments

We present an extensive empirical evaluation of CRTs. Additional details and results for each experiment can be found in the Appendix 9.10.

### 5.1 Comparing variance of CRTs for Gaussian kernel estimation

In this initial experiment, we sample synthetic pairs of vectors $\mathbf{x}, \mathbf{y}$ and evaluate variance of CRTs based on the analytic formulas (6,11,13). Our goal is to check whether there are scenarios when the newly introduced RF mechanisms have smaller variance than existing TrigRF and PosRF methods. We set $d = 64$ which is standard in e.g. Transformer applications (Section 2.3). We use four different regimes for drawing $\mathbf{x}, \mathbf{y}$: `normal` corresponds to $\mathbf{x}, \mathbf{y}$ sampled from $\mathcal{N}(\mathbf{0}_d, \sigma^2\mathbf{I}_d)$, `sphere` corresponds to $\mathbf{x}, \mathbf{y}$ sampled uniformly on a sphere $\sigma\mathcal{S}^{d-1}$, `heterogen` corresponds to $\mathbf{x}$ and $\mathbf{y}$ sampled from two heterogeneous distributions: $\mathcal{N}(\mathbf{0}_d, \sigma^2\mathbf{I}_d)$ and $\mathcal{N}(\sigma\mathbf{1}_d, \sigma^2\mathbf{I}_d)$ and `mnist` corresponds to $\mathbf{x}, \mathbf{y}$ being random images from MNIST dataset [19] resized to $8 \times 8$, scaled by $\sigma > 0$ and flattened.

In many scenarios (see Figure 2), CRTs outperform TrigRF and PosRF baselines. Among other improvements, GERF gives more than $e^{80}, e^{125}, e^{10}$ times variance reduction compared to TrigRF in `normal`, `heterogen` and `mnist` when $\sigma = 1$. OPRF and GeomRF+ give more than $e^{75}, e^{125}, e^7$ times variance reduction compared to PosRF in `normal`, `heterogen` and `mnist` when $\sigma = 1$.

### 5.2 Comparing CRTs in the non-parametric classification

Our next experiment is a non-parametric classification where probabilities are predicted by kernel regression [38, 59] with the Gaussian kernel. Training data consists of objects $\mathbf{o}^{(1)}, \ldots, \mathbf{o}^{(L)} \in \mathbb{R}^d$ with corresponding one-hot encoded labels $\mathbf{r}^{(1)}, \ldots, \mathbf{r}^{(L)} \in \mathbb{R}^n$. The predicted label distribution for the new object $\mathbf{o}^*$ is defined as $\mathbf{r}^* = \sum_{i=1}^L K(\sigma\mathbf{o}^*, \sigma\mathbf{o}^{(i)})\mathbf{r}^{(i)} / \sum_{i=1}^L K(\sigma\mathbf{o}^*, \sigma\mathbf{o}^{(i)})$ where $\sigma > 0$

Table 1: Non-parametric classification, test accuracy (%). $M = 128$. The **best** result, second best.

| Dataset | TrigRF | PosRF | GERF | PoisRF | GeomRF | OPRF | PoisRF+ | GeomRF+ | $L$ |
|---|---|---|---|---|---|---|---|---|---|
| abalone [39] | 12.0 | 16.0 | 17.0 | 18.0 | **18.3** | 17.1 | 14.0 | 15.1 | 3758 |
| banknote [22] | 66.2 | 83.4 | 92.4 | 84.4 | **94.5** | 92.6 | 80.1 | 85.6 | 1233 |
| car [5] | 66.3 | 69.2 | **70.9** | 66.3 | 66.3 | 69.5 | 66.3 | 67.2 | 1554 |
| yeast [30] | 29.7 | 34.4 | 42.9 | 36.9 | 35.9 | **44.4** | 29.7 | 31.0 | 1334 |
| cmc [35] | 46.6 | 45.1 | **47.8** | 46.6 | 47.3 | 46.3 | 35.5 | 43.5 | 1324 |
| nursery [40] | 31.3 | 77.4 | 63.8 | 77.1 | 77.1 | **78.9** | 77.3 | 71.0 | 11664 |
| wifi [47] | 15.2 | 88.8 | 93.3 | 95.3 | **95.8** | 93.3 | 77.2 | 82.9 | 1799 |
| chess [23] | 16.5 | 20.2 | 20.4 | 19.1 | 19.5 | 20.2 | 19.2 | **22.5** | 25249 |
| Average | 35.5 | 54.3 | 56.1 | 55.5 | 56.8 | **57.8** | 49.9 | 52.3 | N/A |

is a hyperparameter tuned on the validation set. Using the RF approximation for the kernel as in (2), we, with $O(nLM)$ preprocessing, can approximate $\mathbf{r}^*$ in $O(nM)$ time per example instead of $O(nL)$ for the exact computation.

Since the predicted class is $\mathrm{argmax}_{1 \leq l \leq n} \mathbf{r}^*$, we can ignore the denominator term and, therefore, use non-positive RFs. We evaluate on classification benchmarks from UCI Repository [24] (Table 1). The best results are achieved by new RF mechanisms, with GeomRF and OPRF performing particularly well. OPRF shows the best average performance, therefore our recommendation for practitioners is to opt for this method. For the same reason, we focus on the FAVOR++ variant (OPRF with orthogonal random projections for attention approximation) in our Transformer experiments below.

### 5.3 FAVOR++ in scalable Transformers

#### 5.3.1 Natural language processing

In this setting, we test different low-rank attention Transformers on the General Language Understanding Evaluation (GLUE) benchmark [57], consisting of 8 different natural language understanding tasks with the sequence length ranging from 32 to 128. We used the same training parameters as mentioned in [20] (see Appendix 9.10.3 for details). We compared FAVOR+ [15], ELU [31] and ReLU [15] variants of the Performers [15] against a FAVOR++ variant and report the results in Table 2. We find that FAVOR++ outperforms all these low-rank Transformers in most GLUE tasks. In particular, FAVOR++ outperforms FAVOR+ on all GLUE tasks, demonstrating downstream effectiveness of the variance reduction of the softmax kernel estimation. Furthermore, warm-starting with pre-trained BERT-base model checkpoint [20] (*Uptrain FAVOR++* in Table 2), further improves performance demonstrating backward-compatibility of FAVOR++ with the exact softmax kernel.

#### 5.3.2 Speech modelling

We compare FAVOR++ with FAVOR+ on speech models with the LibriSpeech ASR corpus ([42]). We apply both to approximate attention blocks in the 17-layer Conformer-Transducer encoder ([26]) of only 4 attention heads and use the word error rate (WER) metric – a standard way to evaluate speech models. In both cases FAVOR++ outperforms FAVOR+, as shown in Figure 3. The WER

Table 2: GLUE Dev results on base sized models. Number of training examples is reported below each task. MCC score is reported for CoLA, F1 score is reported for MRPC, Spearman correlation is reported for STS-B, and accuracy scores are reported for the other tasks. The **best** result, second best.

| System | MNLI 392k | QQP 363k | QNLI 108k | SST-2 67k | CoLA 8.5k | STS-B 5.7k | MRPC 3.5k | RTE 2.5k |
|---|---|---|---|---|---|---|---|---|
| FAVOR+[15] | 80.26 | 89.53 | 87.13 | 90.58 | 53.17 | 85.07 | 83.82 | 67.59 |
| ELU[31] | 80.72 | 90.05 | 89.09 | 91.51 | 48.43 | 86.68 | 85.05 | **68.59** |
| ReLU[15] | 81.39 | 90.11 | 88.85 | 91.97 | 52.08 | **87.64** | 84.56 | 67.51 |
| FAVOR++ | 81.25 | 90.15 | 89.58 | 92.00 | 54.95 | 85.62 | 85.78 | 67.87 |
| Uptrain FAVOR++ | **82.29** | **90.43** | **89.73** | **92.20** | **58.85** | 85.90 | **88.73** | 67.63 |

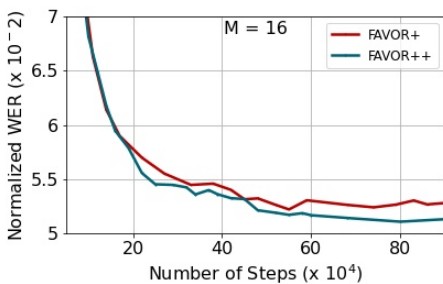 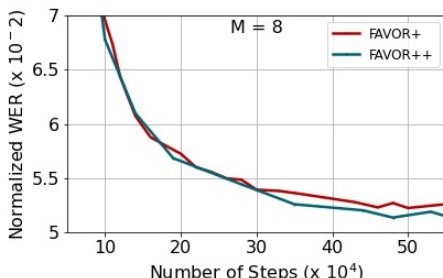

Figure 3: Comparison of the Conformer-Transducer encoder with FAVOR++ and FAVOR+ attention on the LibriSpeech [42] corpus for $M = 16$ and $M = 8$ RFs. We used common word error rate (WER) metric.

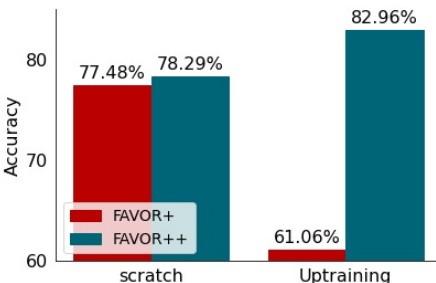 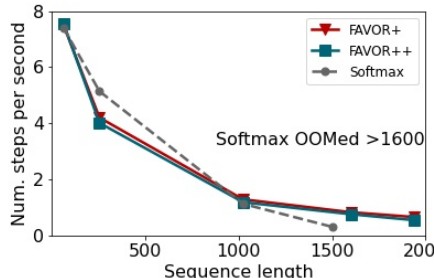

Figure 4: Image-Transformers experiments. **Left:** Accuracy of training FAVOR+ and FAVOR++ on ImageNet from scratch and fine-tuning from softmax MAE pre-trained weights (Uptraining). **Right:** Comparing sequence length vs number of steps per second for FAVOR+, FAVOR++ and regular Transformer variant (Softmax).

improvement for FAVOR++ is substantial: 2.49% for $M = 8$ and 3.05% for $M = 16$ with a negligible $O(Ld) \ll O(LMd)$ overhead for computing (8) compared to FAVOR+.

### 5.3.3 Vision Transformers

To further showcase the need for more accurate softmax kernel approximation, we compare the performance of FAVOR+ and FAVOR++ on ImageNet ([18]). We inject both mechanisms to the attention modules of Vision Transformers (ViT [21]). In Figure 4, we show the results of training from scratch and uptraining from the MAE checkpoint [29]. We see that, as opposed to FAVOR+, FAVOR++ is more stable and is able to improve performance especially for uptraining, demonstrating backward-compatibility with the exact softmax kernel.

Finally, we compare the computational complexity of FAVOR+ and FAVOR++. In Figure 4, the right plot shows the number of steps per second as a function of sequence length $L$ on the same hardware. We see that attention modules using FAVOR+ and FAVOR++ have very similar computation time (both provide linear attention). Moreover, for sequence lengths above $1000$, training a regular ViT model became increasingly difficult due to out-of-memory errors.

## 6 Limitations of this work & broader impact

Several of the mechanisms proposed in this paper can be further extended, potentially leading to even more accurate algorithms. For instance, it remains an open question how to choose theoretically optimal parameters for GERF and GeomRF mechanisms. Furthermore, DIRFs can benefit from optimizing the discrete distributions defining them (to minimize variance of the estimation) rather than choosing them a priori. Our methods should be used responsibly, given rising concerns regarding the carbon footprint of training massive Transformer models and other societal issues [49, 8, 3, 60].

## 7 Conclusion

We presented a new class of RF mechanisms called chefs' random tables (CRTs) including methods providing positivity and boundedness of random features – two key properties for new applications of RFs in Transformer training. We provided comprehensive theoretical results and extensive empirical evaluation, resulting in particular in new state-of-the-art low-rank attention Transformers for text.

## 8 Acknowledgements

V. L. acknowledges support from the Cambridge Trust and DeepMind. V. L. was part-time employed by Google while a PhD student. A.W. acknowledges support from a Turing AI Fellowship under EPSRC grant EP/V025279/1, The Alan Turing Institute, and the Leverhulme Trust via CFI.

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
