# OpenReview forum: "Chefs' Random Tables: Non-Trigonometric Random Features"
_NeurIPS.cc/2022/Conference — NeurIPS 2022 Accept_

### Official Review · Reviewer_xmB9 · 2022-07-09

**Rating:** 6
**Confidence:** 3
**Soundness:** 3 good
**Presentation:** 3 good
**Contribution:** 3 good

**Summary:**


In this paper, the authors propose two alternative schemes of positive random features for Gaussian (Softmax) kernel approximation. i.e., the Generalized exponential RFs and the Discretely-induced RFs.  Variances for both approximations are given.  The authors further show the variance reduction of the OPRFs (a special case of GERF) with orthogonal samples holds for any $d>0$ instead of asymptotically only large $d$ in literature.  A uniform convergence result is proved for (low-rank transformer) attention approximation.

**Questions:**


1. Does the proposed OPRF and discretely-induced RF scheme still construct positve RFs for new data instead of the given data $\boldsymbol{X}$? Note that $\boldsymbol{c}$ and statistics computed from a given set $\boldsymbol{X}$.  They may change for a new data $\boldsymbol{z}$.  As $c_l= min_i min ( \boldsymbol{x}^{(i)} ,\boldsymbol{y}^{(i)}    ) - \epsilon $,  it may be more sensitive for the new data or distribution shift.

2. The variance reduction performance surprised me. However, the discussion in Section 5.1 is quite limited. Could the authors discuss more the experiment in Figure 2? Why the GERF achieves so significant gain?  It seems that the baselines in the top rows are different from that in the bottom.  Could the authors plot them together for better evaluation? Could the authors provide the comparison of variance w.r.t the number of features ($M$)?

**Strengths And Weaknesses:**


Pros.
1.  The proposed OPRF scheme seems promising for constructing positive and bounded random features, which may have the potential for accelerating Transformer while maintaining stable training.

2. In the experiments, the variance reduction is surprisingly significant (up to $e^{60} times).

3.  The paper is well organized and well presented.

Cons.
1.  It seems that the discretely-induced RFs are sensitive to outliers because the requirement $c_l= min_i min ( \boldsymbol{x}^{(i)} ,\boldsymbol{y}^{(i)}    ) - \epsilon $ in Line  181. Moreover, it is not rotation invariant, i.e. the approximation change when the data distribution ratotated.  Note that  Gaussian kernel and softmax kernel are rotation invariant.

2. Some symbols in Theorem are given without definition e.g., $C(\|\boldsymbol{x} + \boldsymbol{y} \|)$  in Theorem 4.1.

---

> ### Author Response · Authors · 2022-08-02
> **Response to Reviewer xmB9**
>
> We would like to sincerely thank the Reviewer for all the comments.
>
> *It seems that the discretely-induced RFs are sensitive to outliers*
>
> Thank you for the comment. While this is a valid concern regarding DIRFs, in the end we have to run an experiment and see which variant performs best in practice. What we find out experimentally is that a) in Figure 2, in `sphere` and `heterogen` regimes, GeomRF+ outperforms other positive-valued variants and b) in Table 1, GeomRF is the second best method after OPRF. Hence, we conclude that there might be scenarios where DIRFs are useful for practitioners. Also, the OPRF method proposed in our paper doesn't raise mentioned concerns.
>
> *Some symbols in Theorem are given without definition e.g., C(|x+y|) in Theorem 4.1.*
>
> The main feature of C in Theorem 4.1 is that it's constant with respect to d, M. For further clarification, we added an explicit definition of C in the proof of Theorem 4.1 in Appendix (lines 614-615).
>
> *Does the proposed OPRF and discretely-induced RF scheme still construct positve RFs for new data instead of the given data X?...*
>
> That is a good point unless the data is given beforehand (as it is in Transformers) or we have prior knowledge that the data is lower-bounded by C. Note that this is not the case for OPRF which is positive-valued for any inputs.
>
> *Could the authors discuss more the experiment in Figure 2?...*
>
> Thank you very much for the comment. Our intuition on why GERF achieves a significant gain in top plots is that GERF generalizes TrigRF, PosRF and OPRF. Though it is not positive-valued as OPRF or PosRF and that is why we plot it in the top row. The bottom row corresponds to positive-valued variants. We plot them separately because positivity is a restriction which might result in a higher variance, but it's important for the application in Transformers. In the revision, we added a plot where all variants are drawn together (Figure 5 in the Appendix).
>
>
> *Could the authors provide the comparison of variance w.r.t the number of features (M)?*
>
> Figure 2 corresponds to M=1. Choosing M > 1 would result in the same analytical variance divided by M for all methods so it's unnecessary.
>
> Here we report ablation accuracy results over M for the kernel classification experiment:
>
> M | TrigRF | PosRF | GERF | PoisRF | GeomRF | OPRF | PoisRF+ | GeomRF+
>
> 16 | 35.5 | 46.9 | 47.4 | 49.0 | **49.1** | 48.5 | 41.3 | 43.2
>
> 32 | 35.5 | 50.5 | 51.2 | 52.2 | **52.6** | 51.8 | 44.1 | 46.5
>
> 64 | 35.5 | 51.3  | 54.0 | 54.2 | 55.2 | **55.4** | 47.0 | 50.0
>
> 128 | 35.5 | 54.3 | 56.1 | 55.5 | 56.8 | **57.8** | 49.9 | 52.3
>
> 256 | 35.5 | 55.6 | 58.1 | 56.5 | 57.8 | **59.7** | 51.9 | 55.0
>
> We see that OPRF consistently outperforms the baselines (TrigRF, PosRF) and also outperforms or is competitive with other methods proposed in the paper. Further, OPRF shows the best performance among positive-valued random features (PosRF, OPRF, PoisRF+, GeomRF+) in all settings. As for the choice of M, we see that performance increases as M grows which is expected. Hence, in practice, a good strategy is to select M as big as the compute budget permits which would alleviate an expensive grid search over M. We added these additional results to the revision (Appendix 8.10.2, Table 4).

---

### Official Review · Reviewer_Csyw · 2022-07-10

**Rating:** 7
**Confidence:** 3
**Soundness:** 4 excellent
**Presentation:** 3 good
**Contribution:** 3 good

**Summary:**

The paper introduces Chefs' Random Tables (CRTs), a family of random features (RFs) for unbiased low-rank approximation of attention matrices in Transformer modules with the ultimate goal of overcoming the quadratic time complexity bottleneck associated to computing full-rank attention matrices. Previous random feature methods for this problem mentioned by the paper have either lacked:
- positiveness [2], resulting in numerical instabilities during the computation of the attention matrix normalization factor (the inverse of the row sums, line 102).
- boundedness or even possessing a finite moment-generating function [3], resulting in high variance estimators and inapplicability of Hoeffding or Chernoff-type inequalities for establishing tail-bounds.

These are the main desiderata that the paper sets out to achieve while maintaining unbiased approximation.

The proposed CRT family encompasses two new classes of random features:

  1) Generalized Exponential Random Features (GERFs), which as the name says, generalizes previous random features defined using exponential functions and Gaussian random projections, and contains as a special case the trigonometric RFs of [1,2] (also often referred to as Random Fourier Features), and the positive RFs of [3] for approximating attention matrices;

2) Discretely Induced Random Features (DIRFs) that lift a discrete probability distribution to a random feature map by making use of the Taylor expansion of the exponential function.

As a special case of 1), the authors introduce Optimal Positive Random Features (OPRFs) by minimizing the random feature variance with respect to the degrees of freedom of the GERF class over (a subset of) their domain. As an instantiation of 2), the authors propose Poisson Random Features (PoisRFs) and Geometric Random Features (GeomRFs).  Explicit formulas are given for computing the variances of each of these techniques.

Additional variance reduction is achieved by replacing the Gaussian random projections in 1) by block-orthogonal ensembles of random projections as done in [2] and [3]. When combining this technique with OPRFs for attention matrix estimation, the resulting methodology is called FAVOR++ for which a variance bound is established in Theorem 4.1.

Further on the theory side, exponential tail bounds are given for OPRFs both with i.i.d. and orthogonal random projections in Theorem 4.2. In Theorem 4.3 a uniform convergence result is given for the first time for approximating attention matrices with positive random features, specifically for the i.i.d. variant of OPRFs.

Experiments are carried out on:
- synthetic data, comparing the variance of the proposed random features with the previous techniques;
- UCI classification problems by using single attention layer as a mapping from input to class membership probability;
- NLP and speech modelling benchmarks;
- Vision Transformers for testing the performance on a long-range dataset.

The experiments overall demonstrate the benefit of some of the proposed features on these kinds of learning problems.

[1] Rahimi, Ali, and Benjamin Recht. "Random features for large-scale kernel machines." Advances in neural information processing systems 20 (2007).

[2] Choromanski, Krzysztof, et al. "Masked Language Modeling for Proteins via Linearly Scalable Long-Context Transformers." arXiv preprint arXiv:2006.03555 (2020).

[3] Choromanski, Krzysztof Marcin, et al. "Rethinking Attention with Performers." International Conference on Learning Representations. 2020.


**Questions:**

Theory:
- In Theorem 3.3, the minimization problem is restricted to the range when $A$ is real and $s = +1$. The authors mention that this results in positive RFs. I wonder if it is possible to get positive RFs outside of this domain, and if so, do the authors think there could be any significantly lower variance estimators achieved by lifting the domain constraint (potentially highlighting future work)?
- If I understand correctly, the $\rho^\star$ computed in eq.(7) is not what is used in practice when a full matrix is required and not only a single kernel evaluation, but the various squared norm terms present in it are replaced with their expectations over the dataset. Does this affect the theoretical analysis given for the OPRFs in Section 4? In particular, if Theorems 4.1 and 4.2 still hold when $\rho$ is not chosen as being optimal for pairs $\mathbf{x}$ and $\mathbf{y}$? For Theorem 4.3, is it assumed that $\rho$ is chosen this way or some other way?

Experiments:
- It was unclear to me which method GERF refers to in Section 5.1 and 5.2. According to Theorem 3.1 and the definition above it, it has two degrees of freedom, $A$ and $s$. Was it mentioned how these were chosen and why?
- In Figure 2, does OPRF refer to the i.i.d. or the orthogonal variant? Regardless, it would be interesting to show both so as to see whether the orthogonal projections really do provide variance reduction in practice as justified theoretically by Theorem 4.1.
-  In Section 5.2, the prediction model used is an attention lookup over the one-hot encoded class labels of training data with testing input as query and training inputs as keys. I understand that the motivation for doing this is to stay within the theme of attention models. I do wonder though if it would be possible to have a proper kernel-based nonparametric model (e.g. SVM or KRR) to demonstrate whether the proposed RFs could be useful for approximating the Gaussian kernel in kernel-based learning tasks? As the theoretical aspect of the paper could be interesting for the wider kernel community, it might be helpful to have some experiments in this direction as well.
- In Sections 5.3.1, 5.3.2 and 5.3.3, I was somewhat missing more alternative methods, since only Performer variants are presented as linear Transformer baselines. There is a wide range of efficient Transformers, and it would be good in my opinion to compare against at least some of these. The optimal method may of course depend on the choice of dataset, see next.
- As it is demonstrated in Figure 4 (Right), full-rank attention is feasible until around 1000 steps, hence the main computational benefits of linear complexity methods only become apparent outside this range. However, although computations can still be feasible, such models can fail to retain long-range interactions. The Long Range Arena benchmark [4] seems to be popular in the community for testing linear complexity Transformers on long-range problems. It would probably increase the impact of the paper if potentially strong results can be achieved on these benchmarks. If it does not perform that well, this experiment could still be of value by potentially pointing out some practical drawbacks of the model, which could further be investigated and explained.

Remarks & typos:
-  Early on in the introduction it was mentioned that positive RFs are required for Transformers as in FAVOR+. The answer to why this is the case was only next provided at the end of page 3. I think it would be useful to have a sentence about why this is needed in the introduction being that it is one of the main motivating factors for the paper as discussed in lines 40-41.
- In Line 59, a $-$ sign is missing within the Gaussian kernel formula

[4] Tay, Yi, et al. "Long Range Arena: A Benchmark for Efficient Transformers." International Conference on Learning Representations. 2020.

**Limitations:**

The authors have adequately discussed this.

**Strengths And Weaknesses:**

The writing of the paper is clear, and in my opinion it reads relatively well as being somewhat well-versed in both random feature kernel approximations and Transformer models. Although the core theme of the paper is not exactly novel, as it is an extension of previous works, the ideas presented are quite original and interesting.

The paper seems quite strong from a theoretical perspective with various results given for the proposed techniques, although I have not checked proofs in detail. The authors claim that so far no uniform guarantees have been established for positive RF based approximation of attention matrices, which according to my knowledge is true. This should make the paper an interesting contribution to theoretical research on attention layers and Transformer modules. Overall, the theoretical aspect is quite sound, although I do have some questions regarding the practicalities (see Questions).

The main weakness of the paper in my opinion is the range of experiments, such as choice of competing methods and lack of long-range datasets. A strong part of the experiments, however, is that it is demonstrated that the proposed RFs allow backwards compatibility for fine-tuning (uptraining) transformer models pretrained with full-rank attention, which should make it possible to scale and extend models pretrained on smaller datasets to much larger datasets by uptraining with the proposed low-rank technique.

To summarize, the paper is quite interesting theoretically, and the experiments support the benefits of the approach, but I do believe it would benefit from some further comparisons to make it more convincing for practitioners, see the Questions section for more details.

---

> ### Author Response · Authors · 2022-08-02
> **Response to Reviewer Csyw, Part 1**
>
> We would like to sincerely thank the Reviewer for all the comments.
>
> *I wonder if it is possible to get positive RFs outside of this domain…*
>
> Thank you very much for the question. While the case of real A, s = 1 is very elegant theoretically and results in substantial improvements in practice, it is unclear how to optimize variance of GERFs (Equation 6) in the most general case and we leave it to future work. Also, it remains unclear whether it is possible to get positive-valued random features when A is complex, since random features become complex-valued and we need to make sure that both real and complex parts have positive signs for all omegas, x and y, which is non-trivial.
>
> *...the various squared norm terms present in it are replaced with their expectations over the dataset. Does this affect the theoretical analysis given for the OPRFs in Section 4?*
>
> Theorem 4.1 only relies on the fact that A is real which always holds for OPRFs even when we use average statistic. Theorems 4.2, 4.3 further rely on the fact that A is a negative real number which also holds when || x + y ||^2 is substituted by its average since the average is positive. We added these clarifications in the revision by slightly modifying Theorem 3.3 and Section 4.
>
> *It was unclear to me which method GERF refers to in Section 5.1 and 5.2.*
>
> GERF refers to a method when A is a complex number and s can be +1 or -1. In this general case, we do not know a closed form optimum for (A, s), hence we use numerical minimization for the variance expression (Equation 6) where we use average statistics (Equation 8) instead of || x + s y ||^2. We mention the details of numerical minimization in Appendix 8.10.1: "We use… two L-BFGS-B routines of 50 iterations to minimize A in GERF for s = -1 and +1 respectively". We added a clarifying sentence about that in the revision (Appendix 8.10.2, Experiment details).
>
> *In Figure 2, does OPRF refer to the i.i.d. or the orthogonal variant?...*
>
> In Figure 2, we compute the variance for M = 1 random feature, so we cannot use orthogonality there. In Table 1, though, we use orthogonal variants for TrigRF, PosRF, GERF and OPRF. We run an additional experiment with non-orthogonal (i.i.d. Gaussian) omegas and obtain the following results (notation: non-orthogonal accuracy / orthogonal accuracy):
>
> Dataset | TrigRF | PosRF | GERF | OPRF
>
> abalone | 12.0 / 12.0 | 15.5 / **16.0** | **17.7** / 17.0 | 16.7 / **17.1**
>
> banknote | 66.2 / 66.2 | **83.9** / 83.4 | 93.2 / **92.4** | 92.3 / **92.6**
>
> car | 66.3 / 66.3 | 68.9 / **69.2** | 70.5 / **70.9** | **69.9** / 69.5
>
> yeast | 29.7 / 29.7 | **34.6** / 34.4 | 42.8 / **42.9** | 42.1 / **44.4**
>
> cmc | 46.6 / 46.6 | 44.7 / **45.1** | 47.4 / **47.8** | **47.3** / 46.3
>
> nursery | 31.3 / 31.3 | 73.2 / **77.4** | 63.8 / 63.8 | 75.8 / **78.9**
>
> wifi | 15.2 / 15.2 | 84.6 / **88.8** | 93.0 / **93.3** | 92.1 / **93.3**
>
> chess | 16.5 / 16.5 | 19.6 / **20.2** | 20.4 / 20.4 | **20.4** / 20.2
>
> Average | 35.5 / 35.5 | 53.1 / **54.3** | 56.1 / 56.1 | 57.1 / **57.8**
>
> We observe that the orthogonal variant does not harm, or improve the results in most cases. Remarkably, two positive-valued random features (PosRF and OPRF) benefit from orthogonality when averaged over all benchmarks. This agrees with our theoretical findings that orthogonal features are more suited for positive-valued features (see Note under Theorem 4.1).  We added these additional results to the revision (Table 6).

---

> > ### Author Response · Authors · 2022-08-02
> > **Response to Reviewer Csyw, Part 2**
> >
> > *I do wonder though if it would be possible to have a proper kernel-based nonparametric model (e.g. SVM or KRR)...*
> >
> > We run an additional experiment and evaluate the kernel ridge regression (KRR) model in the same task of predicting logits on the same set of UCI benchmarks. We follow the kernel classification setup closely with the difference that the model is KRR instead of Nadaraya-Watson kernel regression. For KRR, instead of the $\sigma$ parameter, we tune the ridge parameter $\phi$ on the logarithmic grid of 10 values from 0.01 to 100. We select the best $\phi$ value on the validation performance. The results are as follows:
> >
> > Dataset | TrigRF | PosRF | GERF | PoisRF | GeomRF | OPRF | PoisRF+ | GeomRF+
> >
> > abalone | 10.1 | 21.4 | 21.9 | 23.2 | **25.1** | 21.8 | 16.3 | 13.6
> >
> > banknote | 38.7 | 99.1 | 99.8 | **100.0** | **100.0** | 99.7 | 90.8 | 99.2
> >
> > car | 36.1 | 70.7 | **70.8** | 34.0 | 39.0 | 70.5 | 62.7 | 67.4
> >
> > yeast | 15.9 | 49.2 | 51.8 | 39.4 | 52.2 | **52.6** | 5.2 | 20.8
> >
> > cmc | 34.1 | 46.6 | 47.3 | 44.9 | **49.4** | 47.8 | 37.2 | 40.8
> >
> > nursery | 27.5 | **58.5** | 57.4 | 30.0 | 31.0 | 57.7 | 41.0 | 46.6
> >
> > wifi | 13.7 | 97.2 | **98.1** | 92.7 | 97.1 | 97.3 | 36.1 | 81.6
> >
> > chess | 11.0 | 17.2 | 16.9 | 12.5 | 13.7 | **17.4** | 12.7 | 16.3
> >
> > Average | 23.4 | 57.5 | 58.0 | 47.1 | 50.9 | **58.1** | 37.8 | 48.3
> >
> > We observe that, again, OPRF shows the best average performance which is also slightly better than for the kernel regression model (58.1 against 57.8, Table 1). We added these results into the revised paper (Tables 7, 8).
> >
> > *Comparison to other (than Performers) efficient Transformers in Sections: 5.3.1, 5.3.2 and 5.3.3*,
> >
> > *Long Range Arena*
> >
> > Thank you very much for your suggestions and comments. Following Reviewer’s comment, We run experiments on 3 LRA datasets (ListOps with 2K tokens, Retrieval with 4K tokens, Image with 1K tokens).  We have reported the results in the Appendix: Section 8.11. As we see in Table 18, FAVOR++ improves on the performance of Performer for: ListOps (where the accuracy improves from 36.00 to 42.65) and Retrieval (where accuracy improves from 53.82 to 60.40).
> > On both ListOps and Retrieval datasets we get *the best performance* when compared to *11* other Transformers’ architectures: regular softmax Transformer [55], Synthesizer [52], Sinkhorn [53], Sparse Transformer [10], Reformer [32], Local Attention [43], Longformer [2], Linformer [58], BigBird [64], LinearElu [31] and Performer [15].
> >
> > We also run experiments for the other two LRA tasks, namely: Imdb reviews (4K tokens) and Pathfinder (1K tokens). We did not put the results for these two tasks in Table 18 since we did not manage to reproduce for them the numbers reported in “Long Range Arena: A Benchmark for Efficient Transformers”  (probably due to the hyperparameter setup discrepancy; this was not the case for the other three tasks which are reported in Table 18). We want to emphasize that for these two tasks FAVOR++ provides substantial improvements over Performer.
> >
> > We also would like to note that even though the field of scalable Transformers is an impactful application of our methods, the main topic of this work is not the class of efficient Transformers (nor new efficient attention techniques), but more accurate random feature (RF) methods for approximating Gaussian and softmax kernels. Furthermore, while considering scalable Transformers, the most relevant comparison is with those existing techniques for efficient attention computation that apply random feature map mechanisms for softmax kernel estimation, since our goal is to compare the effectiveness of different RF methods for Gaussian/softmax kernel estimation in downstream tasks. Thus in the experiments with Transformers, we target FAVOR+ which, to the best of our knowledge, is the only existing mechanism applying RFs for softmax kernel estimation (see: our experiments in Section 5.3.1, 5.3.2, 5.3.3). For the convenience of the Reader, we have added comparison with several Performer variants (applying different kernels) to illustrate the importance of the softmax kernel (which, as shown, is in general the most robust and which approximation is the subject of this work) as well as other Transformers (see: the above section regarding new LRA experiments). The critical comparison is the one with Performers applying FAVOR+ method for softmax kernel estimation. All experimental results clearly show that our approximators outperform this method.
> >
> > *I think it would be useful to have a sentence about why this is needed in the introduction being that it is one of the main motivating factors for the paper as discussed in lines 40-41*
> >
> > We add a sentence in lines 35-36 in the revision: "Positive random features guarantee that the denominator in self-attention is a sum of positive numbers, hence it cannot be negative or too small."
> >
> > *In Line 59, a − sign is missing within the Gaussian kernel formula*
> >
> > Fixed in the revision, thank you very much for pointing this out !

---

### Official Review · Reviewer_sgSZ · 2022-07-11

**Rating:** 6
**Confidence:** 2
**Soundness:** 3 good
**Presentation:** 2 fair
**Contribution:** 3 good

**Summary:**

The paper aims at suggesting an effiecent and unbiased estimator of the softmax/Gaussian kernel with positive bounded random features.
Hence, they suggest a mechanism, namely, optimal positive random features (OPRFs)t, and a new set of random features (RF) mechanisms, namely chefs’ random tables, that do not apply trigonometric functions.

Theorems 3.1 and 3.2 by the authors show that they can give a significant variance reduction for Gaussian/softmax kernel estimation by computing the variance of OPRF-based estimators.  Theorem 4.2 by the authors use the boundedness of OPRFs to provide the first exponentially small upper bound for the tails of the Gaussian/softmax kernel estimators reliant on positive RFs.
As a result, the first uniform convergence findings for the softmax attention approximation with positive RFs utilizing OPRFs are given in Theorem 4.3.

In contrast to RKSs (Random Kitchen Sinks - techniques that rely on trigonometric nonlinear mappings to approximate shift-invariant), which only asymptotically reduce variance for sufficiently big dimensionality d, the authors demonstrate that orthogonal random projections paired with OPRFs provably reduce the variance of OPRFs for any dimensionality d (see Theorem 4.1).

Experimental results are also provided.

**Questions:**

Please see my comment regarding the proofs in the appendix. It will be very helpful to explain the derivations done in the proofs.

As I wrote, my only comments are regarding the writing, as someone that is not an expert nor very familiar with the community it was not easy for me, and thus Im very eager to see the other reviewers' comments and authors' responses.



**Limitations:**

I don't consider these as limitations but, as detailed by the authors, it is possible to expand on a few of the techniques discussed in this study, which might result in algorithms that are even more precise.
It is still unclear how to select the theoretically ideal settings for the Generalized exponential RFs system.


**Strengths And Weaknesses:**

Strengths
It seems to be a very novel work with a solid theory, the numerical part is also sufficient.

Weaknesses
My only comments are regarding the writing, as someone that is not an expert nor very familiar with the community it was not easy for me.

1. When you write a theoretical paper (or a paper with theorems and proofs), the theory must be fully detailed. Many derivations are not clearly explained, e.g., in equation 16 in the appendix, the authors should explain that the last derivation holds by opening the term (x+sy)^2 and another, what about the one before?
In (19)- part of Proof of Theorem 3.2, many details are missing explanation, it took so much time to understand why things hold, and still, some things are missing for me. Same for Proof of Theorem 3.5, Proof of Theorem 3.2, and others.

2. The writing in general, is given to an expert in the field, with some additional intuitive explanations, and details - > as someone who is not an expert in this field, it took me too much to understand what is going on, with additional external reading.

---

> ### Author Response · Authors · 2022-08-02
> **Response to Reviewer sgSZ**
>
> We would like to sincerely thank the Reviewer for all the comments.
>
> *Theorem proofs:*
>
> Thanks for your thorough pass through the appendix and for concrete suggestions on presentation improvement. We have significantly changed the proofs of Theorems 3.1, 3.2, 3.4, 3.5 in the revision by adding many clarifications and de-densifying chains of equations. We hope this addresses the concern about readability. We would be happy to address more comments/ideas related to clarity of proofs during the discussion period.
>
> *Writing … given to an expert in the field:*
>
> Thank you very much for the comment. In the camera-ready version we will de-densify the paper and add paragraphs providing additional intuition, including: (a) explanation why positive random features are particularly useful for training implicit low-rank-attention Transformer models as well as: (b) sketches of the proofs listing the main conceptual ideas.
>
> *Possible extensions of the presented results:*
>
> Thank you for the comment. We believe that our paper leads to several new exciting questions, in particular regarding the optimal variants of the positive random feature mechanisms, as mentioned by the Reviewer and explained in the paper. This undoubtedly will be the subject of our future work.

---

### Official Review · Reviewer_JjGy · 2022-07-12

**Rating:** 5
**Confidence:** 3
**Soundness:** 3 good
**Presentation:** 2 fair
**Contribution:** 2 fair

**Summary:**

The paper proposes a novel class of random features and provides concentration bounds on the approximation properties for the resulting kernel matrices. The main motivation is to have positive random features because the classical Fourier features have not been performing well in Transformers due the fact that they can take negative values. The latter insight comes from prior work and it could have been discussed in more details here.

**Questions:**

lines 87-90: when reviewing the features for positive RFs, can you please comment on the resulting kernel that they approximate? The motivating example started with a Gaussian kernel but positive features do not appear to approximate it (or at least this is not trivial to see).

Is there any difference to approximation or numerical properties of random features from Eq. (4) for different values of constants? What is the significance of Theorem 3.1 in this regard?

In experiments for Figure 2, can you clarify how the variance is computed? Is this at the level of Gaussian kernel or the softmax kernel in Transformers?

 How numerically stable are features in Section 3.2.2 (geometric) given that there is factorial in the denominator?

 Could you review arguments from prior work that would motivate positive random features? It is not trivial to comprehend why such a constraint would be instrumental for the approximation scheme?

 Why is FAVOR+ failing dramatically on "Uptraining", Figure 4?


**Limitations:**

No ethical concerns.

**Strengths And Weaknesses:**

#### - originality
The work builds on existing insights and aims at alleviating some issues in approximating self-attention operators in Transformers using random features. The theoretical results appear to be novel and original, though.

#### - quality
It is theoretically sound paper with bounds on the approximation properties of a novel class of random features. It is difficult to tell how sharp these bounds are, (e.g., Theorems 4.2 and 4.3) but the empirical results indicate significant performance improvements over baselines.
A detailed ablation study that would contrast classical random features to the proposed ones relative to a number of hyper-parameters would be useful and it is missing. For instance, I do not see why would one have more benign learning setting in linear regression with random features using the positive variant.

#### - clarity
The paper is mostly clear, but lacks more detailed references to results from prior work. It builds strongly on FAVOR+ algorithm and findings from that paper. However, the review of that work is limited and one would need to read it to completely grasp the motivation.

#### - significance
The approach can bring speeds up when computing self-attention mechanisms and appears to have better theoretical and empirical properties compared to prior FAVOR algorithm. However, this improvement is mostly incremental and is likely to have a limited impact on this class of models.

---

> ### Author Response · Authors · 2022-08-02
> **Response to Reviewer JjGy, Part 1**
>
> We would like to sincerely thank the Reviewer for all the comments.
>
> *Discussion of the  prior work regarding poor performance of the original Fourier random features (RFs) for Transformers and the need of positive RFs:*
>
> *Could you review arguments from prior work that would motivate positive random features?*
>
> Thank you very much for the comment. To the best of our knowledge, this observation was established in [13], which we refer to several times in the manuscript (as early as in l.5 of the Abstract). Furthermore, in l.100-104 we explain why positive RFs are particularly useful for Transformer training. In the Transformer setting, very often many attention-matrix entries are close to zero. If RF-mechanisms that do not provide positiveness are applied, they can lead to approximators taking negative values for such small entries. This might lead in principle to approximate partition functions (defined as the sums of attention-matrix entries in rows), used to renormalize attention, taking negative values and consequently very unstable training.
>
> The experimental analysis of stability of trigonometric random features is done in [13] – compare training curves in Figure 5 (right), training with trigonometric features is very unstable. Also, see Figure 16 (left), training with trigonometric features ("cos") halts with NaNs very quickly.
>
> Following Reviewer’s suggestion, in the camera-ready version we will extend the l.100-l.104 text-block to a full paragraph, providing  additional details and intuition.
>
> *Detailed ablation study that would contrast classical random features to the proposed ones relative to a number of hyper-parameters would be useful:*
>
> Note that the optimal parameters of OPRF (FAVOR++ in the context of Transformers) can be computed in O(1) time (equation 7) and don't require grid search. The main hyperparameter is M, the number of random features. Here we report ablation results over M for the kernel classification experiment:
>
> M | TrigRF | PosRF | GERF | PoisRF | GeomRF | OPRF | PoisRF+ | GeomRF+
>
> 16 | 35.5 | 46.9 | 47.4 | 49.0 | **49.1** | 48.5 | 41.3 | 43.2
>
> 32 | 35.5 | 50.5 | 51.2 | 52.2 | **52.6** | 51.8 | 44.1 | 46.5
>
> 64 | 35.5 | 51.3  | 54.0 | 54.2 | 55.2 | **55.4** | 47.0 | 50.0
>
> 128 | 35.5 | 54.3 | 56.1 | 55.5 | 56.8 | **57.8** | 49.9 | 52.3
>
> 256 | 35.5 | 55.6 | 58.1 | 56.5 | 57.8 | **59.7** | 51.9 | 55.0
>
> We see that OPRF consistently outperforms the baselines (TrigRF, PosRF) and also outperforms or is competitive with other methods proposed in the paper. Further, OPRF shows the best performance among positive-valued random features (PosRF, OPRF, PoisRF+, GeomRF+) in all settings. As for the choice of M, we see that performance increases as M grows which is expected. Hence, in practice, a good strategy is to select M as big as the compute budget permits which would alleviate an expensive grid search over M. We added these additional results to the revision (Appendix 8.10.2, Table 4).
>
> *Explanation of the  benign learning setting in linear regression with random features using the positive variant:*
>
> Thank you for the comment. Good performance of OPRF in the benign kernel classification experiment can be explained not only by positivity, but by the fact that it results in the smallest variance of the approximation. Here we compute the average log-variance of the approximation among all 8 UCI benchmarks we use in the paper:
>
> TrigRF | PosRF | GERF | PoisRF | GeomRF | OPRF | PoisRF+ | GeomRF+
>
> -0.8 | -0.0 | -20.5 | -0.7 | -7.0 | -20.5 | 36.2 | -16.7
>
> We observe that OPRF has the smallest variance with the same value only for GERF (which can be explained since GERF extends OPRF but is complex-valued, hence we need to use 2 times less features for the comparable amount of computation). The smallest variance for OPRF can be explained by the input data distribution due to which OPRF's variance (first equation in the proof of Theorem 8.4) is smaller in average than other variances (Section 2.2 and Equations 11, 13). We added these additional results to the revision (Appendix 8.10.2, Table 5).

---

> > ### Author Response · Authors · 2022-08-02
> > **Response to Reviewer JjGy, Part 2**
> >
> > *References to prior results:*
> >
> > We have cited in the paper several works on random features, in particular the trilogy of the foundational papers of the field ([30,40,41], l.20). Altogether, we cite 17+ papers on random features, including theoretical results (e.g. the quality of RF-based approximation for kernel regression, QMC variants of RFs and more) and practical ones (e.g. random vs Nystrom features). Following Reviewer’s comments, we will include additional references with discussion, in particular:
> >
> > * Random Feature Attention, Peng et al., ICLR 2021,
> > * Implicit Kernel Attention, Song et al., AAAI 2021,
> > * Random Features for Kernel Approximation: A Survey on Algorithms, Theory, and Beyond, Liu et al. 2021.
> >
> > *Incremental improvement as compared to FAVOR:*
> >
> > We emphasize that our new RF-mechanisms can be applied beyond the FAVOR setting (implicit Transformers), in particular whenever RFs for a Gaussian/softmax kernel are applied. This includes (in addition to scalable Transformers): kernel ridge regression, SVM methods, Gaussian processes, Predictive State Recurrent Neural Networks and more.
> >
> > *...can you please comment on the resulting kernel that they approximate?*
> >
> > All random features discussed in the paper approximate the Gaussian kernel – we mention that in line 82 for TrigRFs and PosRFs, line 114 for GERFs and line 152 for DIRFs.
> >
> > *Is there any difference to approximation or numerical properties of random features from Eq. (4) for different values of constants?*
> >
> > Arbitrary values of A, B, C, D, s may not result in the approximation of the Gaussian kernel. Theorem 3.1 gives the conditions when GERFs are an unbiased approximation of the Gaussian kernel. As we observe in Theorem 3.1, B, C, D can be expressed through A, s which are free parameters. Different values of A, s result in a different variance of the estimator and also whether the features are positive/non-positive and bounded/unbounded. We optimize A, s to reduce the variance while the approximation remains unbiased and features are positive-valued. For the optimum (OPRF) we find that random features are also bounded which is an important property to get tight concentration results (Theorem 4.2).
> >
> > *In experiments for Figure 2, can you clarify how the variance is computed? Is this at the level of Gaussian kernel or the softmax kernel in Transformers?*
> >
> > Figure 2 illustrates variance for the Gaussian kernel estimation – we emphasized that in line 217 in the revision. The variance for a given pair of x and y is computed based on the analytic formulas (Equations 6, 11, 13).
> >
> > *How numerically stable are features in Section 3.2.2 (geometric) given that there is factorial in the denominator?*
> >
> > We normalize features so that their magnitude is bounded by one by first computing everything in logarithms and then subtracting the maximal random feature value. This doesn't affect computations since the same random features appear in the numerator and denominator. We use `scipy.special.loggamma` for computing factorial logarithms, analogous functions exist in the deep learning software (Tensorflow, Pytorch, JAX).
> >
> > *Why is FAVOR+ failing dramatically on "Uptraining", Figure 4?*
> >
> > Thank you very much for the comment. We believe the performance gap in the Uptraining setting (Figure 4) is due to the lower variance of FAVOR++ compared to FAVOR+. Low variance is more important in the Uptraining scenario compared to Scratch, since the approximation should be backward compatible with exact attention. This explains why the gap is bigger than in the first scenario (Scratch).

---

### Author Response · Authors · 2022-08-02
**Response to all Reviewers**

We appreciate all the Reviewers’ time and comments. We have provided a response to all questions from each Reviewer. In line with our responses, we have updated our paper in the new revision on OpenReview. All new changes are temporarily highlighted in violet for convenience. Here we summarize the main changes:

* Presentation improvements/typo fixes in the Introduction, Sections 2.1, 3.1, 4, 5.1.

* Proofs of Theorems 3.1, 3.2, 3.4, 3.5 are rewritten with many clarifications added and equations de-densified.

* New experimental results, ablation study and clarifications are added in Appendices 8.10.1, 8.10.2.

* Comparisons with efficient transformers on Long Range Arena tasks with sequences of length up to 4K added in Appendix 8.11.

---

### Author Response · Authors · 2022-08-08
**Message to all Reviewers**

We thank you again for your time and helpful comments. We hope we have addressed your concerns and you might consider raising your score. If you have any further concerns, we'd be glad to address them.

---

### Meta-Review · Area_Chair_JehX · 2022-08-20

**Recommendation:** Accept
**Confidence:** Certain

**Metareview:**

The primary motivation of the paper is the scalable training of transformers, particularly their efficient softmax-attention approximation (3). As a classical approach relying on trigonometric random Fourier features (RFF) does not guarantee positivity (which makes the training of transformers unstable), the authors consider non-trigonometric RFFs. Particularly, they propose a GERF (generalized exponential random features, specified in (4)) for the approximation of the Gaussian kernel (and the softmax kernel) which beyond the previously designed positivity of RFFs [15], can also ensure the boundedness of the random features. They analyze when these RFs give rise to unbiased kernel approximation (Theorem 3.1), establish their variance (Theorem 3.2 for fixed (x,y) inputs), and restricting its free parameters (A to be real, s = 1) they specialize the design to the minimum variance estimator referred to as OPRF (optimal positive random feature). They show tail bounds (Theorem 4.2) and attention approximation guarantees for OPRFs. They also present a DIRF (discretely-induced random features) construction with focus on the Poisson and Geometric designs, which with a shift (Section 3.2.3) can be turned to be positive. The resulting CRT (chefs' random tables) kernel approximation family is illustrated in classification (on UCI benchmarks) and in training transformers (in NLP, audio and image context).

Kernel methods are without doubt at the forefront of machine learning; developing new kernel approximation schemes is of significant interest to the NeurIPS community. As it was assessed by the reviewers, the authors present novel and valuable theoretical insights in this respect, with convincing numerical illustrations. As the reviewers pointed out the manuscript could be improved somewhat by (i) more detailed references to results from prior work, and (ii) making it more accessible to wider audience. Please incorporate these comments in the final version of the manuscript.

**Award:**

No

---

### Decision · Program_Chairs · 2022-09-14

Accept